# FP4 All the Way: Fully Quantized Training of LLMs

**Brian Chmiel** [^,*,†] **Maxim Fishman** [^,*,†] **Ron Banner** [^,†] **Daniel Soudry** [°]

[^]Nvidia, Israel
[°]Department of Electrical and Computer Engineering - Technion, Haifa, Israel

{brianchmiel, maxim.fishman.d, ronbanner.RB, daniel.soudry}@gmail.com

## Abstract

We demonstrate, for the first time, fully quantized training (FQT) of large language models (LLMs) using predominantly 4-bit floating-point (FP4) precision for weights, activations, and gradients on datasets up to 1T tokens. We extensively investigate key design choices for FP4, including block sizes, scaling formats, and rounding methods. Our analysis shows that the NVFP4 format, where each block of 16 FP4 values (E2M1) shares a scale represented in E4M3, provides optimal results. We use stochastic rounding for backward and update passes and round-to-nearest for the forward pass to enhance stability. Additionally, we identify a theoretical and empirical threshold for effective quantized training: when the gradient norm falls below approximately $\sqrt{3}$ times the quantization noise, quantized training becomes less effective. Leveraging these insights, we successfully train a 7-billion-parameter model on 256 Intel Gaudi2 accelerators. The resulting FP4-trained model achieves downstream task performance comparable to a standard BF16 baseline, confirming that FP4 training is a practical and highly efficient approach for large-scale LLM training. A reference implementation is supplied in https://github.com/Anonymous1252022/fp4-all-the-way.

## 1 Introduction

The rapid advancement of Large Language Models (LLMs) has led to unprecedented breakthroughs in natural language understanding and generation. State-of-the-art models now scale to hundreds of billions of parameters, enabling remarkable capabilities across diverse applications. However, this progress comes at a significant cost, with training and inference demanding immense computational power and memory bandwidth. As model sizes grow, hardware constraints become a major bottleneck, necessitating innovations in numerical precision and memory-efficient architectures.

Until recently, the dominant numerical format for pretraining Large Language Models (LLMs) was BF16, which provided a balance between precision and efficiency. However, as model sizes and dataset scales have grown, researchers have explored lower-precision alternatives to improve computational efficiency and reduce memory requirements. A few pioneering studies have demonstrated that full training in FP8 is not only feasible but also effective. [13] showcased the potential of FP8 training on small-scale datasets (100B),[8] extended it to trillions of tokens dataset with a 7B parameters model, and [6] demonstrated FP8's viability at an even larger scale, successfully training a massive 671B parameter Mixture-of-Experts (MoE) model on a vast dataset, achieving state-of-the-art results. FP8 is quickly emerging as the new standard for large-scale LLM training. As research continues to push precision boundaries further, the next logical step is exploring FP4, which promises even greater efficiency while maintaining training stability and model accuracy.

---

[^*]Equal contribution
[^†]Work done while at Intel.

39th Conference on Neural Information Processing Systems (NeurIPS 2025).

Recently, NVIDIA introduced its new GPU architecture, Blackwell [1], marking a major milestone as the first GPU to support FP4 matrix multiplication in hardware. This advancement enables nearly 2× acceleration in matrix multiplication throughput compared to FP8, significantly boosting performance and energy efficiency for large-scale deep learning workloads. Blackwell supports two distinct FP4 formats—MXFP4 and NVFP4—each with different design choices in block size and scale encoding. In this paper, we aim to systematically investigate the full range of block sizes and scaling formats for FP4 training, analyzing their impact on accuracy and stability and highlighting the advantages of the NVFP4 format.

Beyond the choice of numerical format and block size, another critical aspect of low-precision quantization is the rounding mode used during quantization. The two most commonly used rounding modes are round-to-nearest (RtN), which deterministically maps each value to its closest representable value, and stochastic rounding (SR), which probabilistically rounds based on the distance to neighboring values to reduce quantization bias. While prior empirical work [19, 4] has highlighted the importance of SR, particularly in the backward pass for stabilizing gradient updates, its benefits have largely been observed heuristically. In this work, we provide an analysis of noisy gradients training and identify the point where they stop being effective and show the importance of SR in low-precision training.

Recent studies have explored the use of FP4 formats for training large language models, yet none achieved full quantization across all key components—weights, activations, and gradients. In contrast, our approach introduces, for the first time, a fully quantized FP4 training framework that addresses all these components simultaneously, and on significantly larger datasets. Table 2 highlights these important distinction. This end-to-end FP4 training setup enables us to evaluate the practical viability of low-precision methods at scale. Moreover, we evaluate our FP4 models on various downstream tasks—showing on-par results with the BF16 baseline—demonstrating that efficiency gains do not come at the cost of quality.

We make several key contributions:

1. **FP4 Format Optimization.** We conduct comprehensive experiments on block size and scaling formats for FP4 training, revealing that block sizes below 16 elements offer diminishing returns on accuracy. Our comparison of different exponent-mantissa configurations (E1M6 through E8M0) found the best performance was achieved by E4M3—the format used in NVFP4, validating NVIDIA's hardware design choices.

2. **Split Rounding Strategy.** We establish that combining stochastic rounding in the backward pass with round-to-nearest in the forward pass significantly improves training stability and final model accuracy in FP4 training compared to using either method alone throughout the training process.

3. **Precision Transition Analysis.** We provide a theoretical framework that identifies the critical point at which FP4 precision becomes less effective for continued training progress. Specifically, when the full-precision gradient standard deviation falls below $\sqrt{3}$ times the quantization noise standard deviation. We suggest, at the end of the training, to apply a higher precision Quantization Aware Finetuning (QAF) to increase the signal-to-noise ratio higher above this threshold and quickly converge to the baseline.

4. **End-to-End Large-Scale FP4 Training.** We successfully train a 7B-parameter LLM entirely in FP4 precision on a trillion tokens, using 256 Intel Gaudi2 accelerators. While a slight gap in final training loss is observed compared to the BF16 baseline, it is fully closed through the brief QAF phase. This leads to downstream task performance on par with BF16, confirming the practical viability of FP4 for real-world large-scale applications.

## 2 Related work

Quantization is a key area of research in the effort to compress neural networks for more efficient deployment and reduced resource consumption. The two predominant approaches in this space are Post-Training Quantization (PTQ) [22, 11, 9] and Quantization-Aware Training (QAT) [7, 3, 12]. PTQ focuses on converting pre-trained models to low-bit representations without additional training, making it attractive for rapid deployment, especially in inference scenarios. In contrast, QAT incorporates quantization effects during model training or fine-tuning, enabling the network to adapt and maintain accuracy under low-precision constraints. Both methods have seen significant

advancements, with recent work demonstrating competitive performance at 4-bit precision [9] and below [20].

Fully Quantized Training (FQT) is a more challenging task than PTQ or QAT, as it requires training from scratch with low-precision weights, activations, and gradients to accelerate all matrix multiplications. Until recently, applying FQT beyond 16-bit precision was considered difficult due to instability and convergence issues. However, recent works have demonstrated its feasibility. [13] presented the first FQT of a large language model in FP8 on a dataset of up to 100 billion tokens. [8] extended this to 2 trillion tokens, revealing stability issues in later training stages and proposing a modified activation function to address them. [6] further advanced the field by training a large Mixture-of-Experts (MoE) model with FP8 FQT, mitigating instability through finer-grained quantization.

Finer granularity quantization is emerging as a key direction for enabling Fully Quantized Training (FQT) beyond FP8, particularly in the context of FP4 precision. By reducing the quantization block size, these methods aim to better capture local variations in data distributions, improving stability and accuracy. Notable examples include MXFP4 [15] and NVFP4 [1].

The two works most closely related to ours are [21, 19]. [21] proposes training large language models using a vector-wise FP4 format combined with two key techniques: a Differentiable Gradient Estimator (DGE) to replace the standard Straight-Through Estimator (STE), and Outlier Clamp Compensation (OCC) to handle activation outliers wih an additional sparse residual matrix. Their models are trained on up to 100 billion tokens, but they quantize only weights and activations, keeping gradients in higher precision—thus accelerating only one of the three matrix multiplications involved in training. [19], in contrast, focuses on gradient quantization using the MXFP4 format and applies stochastic rounding alongside the Hadamard transform to stabilize training. They train models up to 40 billion tokens. However, like [21], they only accelerate part of the three matrix multiplications. In contrast, our work is the first to demonstrate full FP4 Fully Quantized Training (FQT) of large-scale LLMs, enabling the acceleration of all matrix multiplications during training.

# 3   FP4 training

Going beyond FP8 to FP4 training presents significant challenges due to the limited dynamic range of FP4, making it difficult to capture the full variability of activations and gradients without excessive quantization error. However, the recently introduced microscaling floating-point family (MXFP) [15] offers a promising alternative by dynamically adjusting the scale at finer granularity, mitigating precision loss.

MXFP4, includes 1 sign bit, 2 exponent bits, and 1 mantissa bit (E2M1) is a floating-point format that enhances low-precision training by dividing data into blocks of size 32, with each block sharing a common scale. The scale for each block is stored using the E8M0 format, an 8-bit exponent-only representation that provides a wide dynamic range without a mantissa and sign. Another potential format for FP4 training is NVFP4, which uses the same E2M1 data representation as MXFP4 but differs in block size and scaling format. NVFP4 divides data into smaller blocks of size 16, compared to MXFP4's 32, allowing for finer-grained scaling adjustments. Additionally, NVFP4 employs an E4M3 format for storing scales, providing a balance between dynamic range and precision. Both MXFP4 and NVFP4 are supported in NVIDIA's Blackwell architecture [1]. In Table 1 we compare these 2 formats.

Table 1: Comparison of MXFP4 and NVFP4 Formats

| Datatype | MXFP4 | NVFP4 |
|---|---|---|
| Data Representation | E2M1 | E2M1 |
| Block Size | 32 | 16 |
| Scale Format | E8M0 | E4M3 |
| Per-Tensor Scale | No | Yes |

### 3.1 Exploring block size and scale format

Seeing the differences between the two FP4 formats supported in Blackwell—MXFP4 and NVFP4—in terms of block size and scale format, we decided to investigate their impact further. Specifically, we aim to compare the full range of possible scale formats, while maintaining the FP8 data format. Additionally, we explore different block sizes to understand their effect on numerical stability, training efficiency, and model accuracy. This analysis will provide deeper insights into the trade-offs between dynamic range, precision, and computational efficiency in low-precision training.

In Fig. 1 we train a 350M Llama-style model with FP4 format (E2M1) with block size 16 and different scaling formats. Note that, similar to NVFP4, most configurations (except for E8M0, which corresponds to MXFP4) do not utilize the sign bit in the scale. This may represent a potential inefficiency that future work could aim to exploit. We notice that the best results are achieved with E3M4 and E4M3, where the latter is used in the NVFP4 format. In Fig. 2 we compare different block sizes both with scales formats E8M0 and E4M3, which are the scales used in MXFP4 and NVFP4, respectively. Note that while selecting the appropriate scale has a significant impact on final accuracy—for example, E1M6 leads to complete divergence—the block size has a more modest effect, with smaller block sizes generally yielding better results. This leads us to proceed with the NVFP4 format, which uses a block size of 16 and a scale format of E4M3.

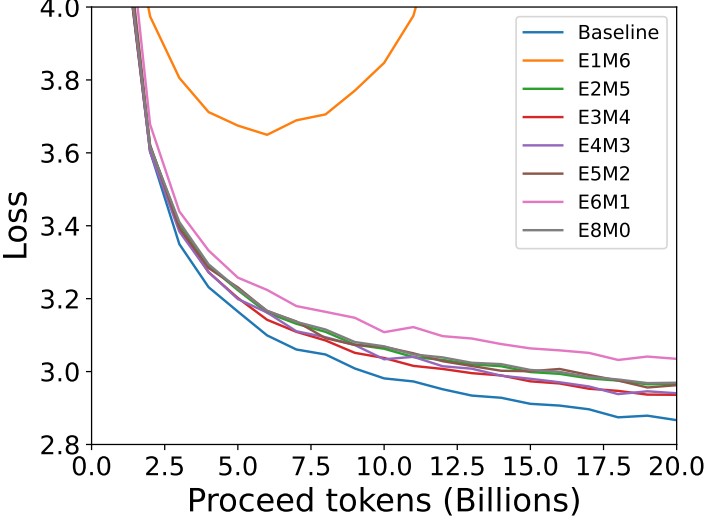

Figure 1: **Formats E4M3 (used in NVFP4) and E3M4 achieved the best results.** Comparison of different scaling formats (E1M6, E2M5, E3M4, E4M3, E5M2, E6M1, E8M0) when training a 350M Llama model using FP4 format (E2M1) with block size 16. The formats E3M4 and E4M3 achieve the best results (recall E4M3 is used in NVFP4), whereas E1M6 results in complete divergence.

### 3.2 Exploring the rounding modes

After completing our exploration of the various block sizes and scales within the FP4 format, we now turn our attention to another critical aspect: the choice of rounding modes. In the next section, we will delve into the different rounding strategies available for FP4 and analyze their impact on numerical stability and model performance.

Fully quantized training encompasses the quantization of three key general-matrix-multiplications (GEMMs): forward, backward, and update. Each GEMM involves two quantized operands—resulting in six distinct quantization points across the training pipeline:

$$\textbf{[Forward]} \quad z_l = Q(W_l)Q(a_{l-1}); \qquad a_l = f_l(z_l) \tag{1}$$

$$\textbf{[Backward]} \quad g_{l-1} = Q(W_l^T)Q(\delta_l); \quad \delta_l = f_l'(z_l) \odot g_l \tag{2}$$

$$\textbf{[Update]} \quad \frac{\partial C}{\partial W_l} = Q(\delta_l)Q(a_{l-1}^T), \tag{3}$$

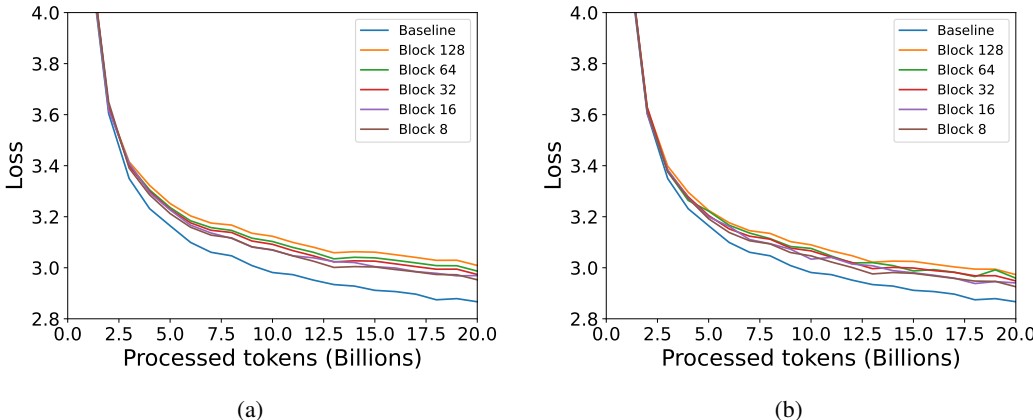

(a)                            (b)

Figure 2: **Block size 16 is the best option.** We examine the impact of different block sizes (8, 16, 32, 64, 128) on training accuracy using scaling formats: (a) E8M0 (used in MXFP4) and (b) E4M3 (used in NVFP4). Smaller block sizes yield modest improvements in accuracy, with diminishing returns below 16 elements per block. Thus, a block size of 16 provides an optimal compromise between performance and computational overhead.

where $C$ is the loss function, $Q$ is a quantization operation, $\odot$ is a component-wise product and, in each layer $l$, $f_l$ is the activation function, $W_l$ is weight matrix, $z_l$ are the pre-activations, and $g_l \triangleq \frac{\partial C}{\partial a_l}$.

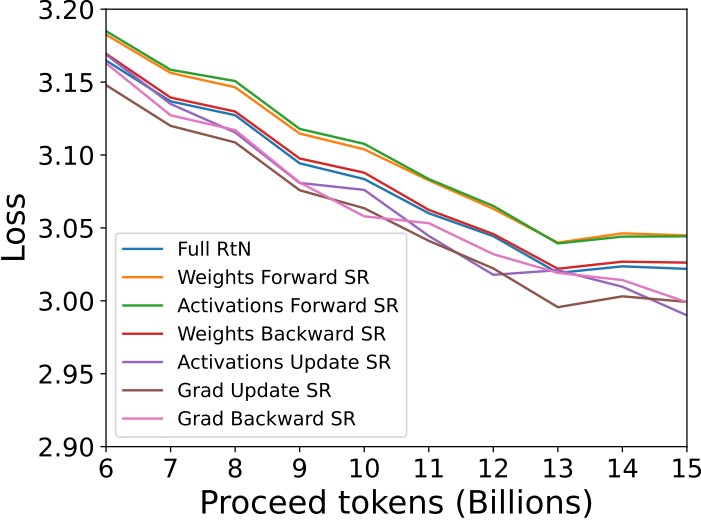

Figure 3: **Comparison of different rounding schemes when training a 350M Llama model using NVFP4 format.** In each graph, we apply SR in one of the six elements in one of the GEMMs while the rest use round-to-nearest (RtN). Notice that applying SR to neural gradients during both 'Update' and 'Backward' GEMMs and activations during the 'Update' GEMM leads to lower training loss, while applying SR to other components has the opposite effect, increasing the loss.

Notably, we have the flexibility to select the rounding mode independently for each of these six elements. In Fig. 3, we present results of training a 350M Llama model using NVFP4 where stochastic rounding (SR) is applied to each of these elements separately while the rest of the elements use round-to-nearest (RtN), allowing us to evaluate its individual contribution. Applying SR to neural gradients during update and backward GEMMs and activations during the update GEMM helps reduce training loss, whereas using SR in other components leads to an increase in loss. In Fig. 7 in the Appendix, we present additional experiments that support the same conclusion. As a result, we

adopt the following selective rounding scheme:

$$\textbf{[Forward]} \quad z_l = Q_{\text{RtN}}(W_l) Q_{\text{RtN}}(a_{l-1}); \qquad a_l = f_l(z_l) \tag{4}$$

$$\textbf{[Backward]} \quad g_{l-1} = Q_{\text{RtN}}(W_l^T) Q_{\text{SR}}(\delta_l); \quad \delta_l = f_l'(z_l) \odot g_l \tag{5}$$

$$\textbf{[Update]} \quad \frac{\partial C}{\partial W_l} = Q_{\text{SR}}(\delta_l) Q_{\text{SR}}(a_{l-1}^T), \tag{6}$$

# 4 Analysis of Quantized SGD with Stochastic Rounding

This section analyzes when training with low-precision gradients stops being effective when using stochastic rounding (SR). While SR removes bias and enables stable descent during much of training, its benefits diminish once gradients become too small relative to quantization noise. We derive a threshold on the gradient-to-noise ratio that signals when further progress stalls, motivating a precision switch for backward and update passes. This switch improves convergence without altering the forward pass or the deployed model.

**Key takeaways from the analysis:**

- **SR enables unbiased updates**, allowing stable descent even under aggressive FP4 quantization. In contrast, deterministic rounding introduces a persistent bias, leading to an irreducible error floor and preventing convergence (see Appendix B.2).
- **There exists a critical threshold:** With SR, we show in Section 4.1 (and in more detail in Appendix B.1) that the average per-coordinate gradient magnitude falls approximately below $\sqrt{3}$ times the quantization noise standard deviation, training no longer yields effective loss reduction. This threshold is derived under simplifying assumptions—e.g., gradient descent with optimal step size, Taylor approximation of the loss, and a concentrated Hessian spectrum.
- **Empirical evidence supports the theory:** in Section 4.2, for both synthetic and real-model settings, we show empirically performance degrades sharply below this threshold. A precision switch guided by the theory restores convergence.

## 4.1 Theoretical Derivation

We begin with a second-order Taylor expansion of the loss function around the current parameter vector $\theta_t$, which reveals a descent term proportional to $-\nabla L^T \Delta\theta$ and a curvature term involving the Hessian $H$. We then replace the full-precision gradient $\nabla L$ with its quantized version $g_q = \nabla L + \varepsilon$, where $\varepsilon$ is zero-mean noise introduced by stochastic rounding.

Using the update rule $\Delta\theta = -\eta g_q$ and taking expectations under SR ($\mathbb{E}[\varepsilon] = 0$, $\mathbb{E}[\varepsilon\varepsilon^T] = \sigma_q^2 I$), we obtain the expected loss change:

$$\mathbb{E}[\Delta L] = -\eta \|\nabla L\|^2 + \tfrac{1}{2}\eta^2 \left( \nabla L^T H \nabla L + \sigma_q^2 \operatorname{tr}(H) \right).$$

Balancing the descent and noise terms yields the optimal step size:

$$\eta^* = \frac{\|\nabla L\|^2}{\nabla L^T H \nabla L + \sigma_q^2 \operatorname{tr}(H)}.$$

Substituting $\eta^*$ back into the expected loss gives:

$$\mathbb{E}[\Delta L] = -\frac{\|\nabla L\|^4}{2 \left( \nabla L^T H \nabla L + \sigma_q^2 \operatorname{tr}(H) \right)}.$$

Differentiating with respect to $\sigma_q$, and using some simplifying assumptions reveals that sensitivity to quantization noise peaks when:

$$\sigma_{\text{critical}} = \frac{\|\nabla L\|}{\sqrt{3d}}.$$

Once the per-coordinate gradient magnitude drops below $\sqrt{3}\,\sigma_q$, the descent becomes negligible and higher-precision gradients are needed to continue improving the loss.

## 4.2 Empirical Validation

To validate this theoretical threshold, we present two types of experiments.

First, we simulate training on a simple quadratic loss with an adaptive noise schedule. We scale the quantization noise to $\sigma_q = k \cdot \sigma_{\text{critical}}$ for $k = 2, 1, 0.5$. As shown in Fig. 4, convergence completely stalls at high noise levels (e.g., $k = 2$), slows near the critical threshold ($k = 1$), and closely tracks full-precision training when noise is reduced below the threshold ($k = 0.5$).

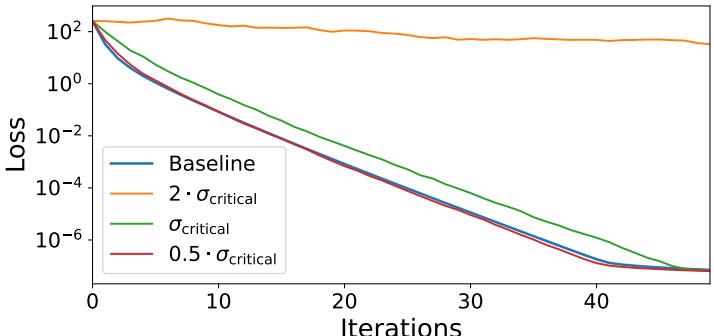

Figure 4: **Validation of theoretical derivation in a simple quadratic loss.** Training loss with noise levels $\sigma_q = k \cdot \sigma_{\text{crit}}$ for $k = 2, 1, 0.5$ in a toy quadratic model. High noise blocks descent; low noise allows continued progress.

Second, we test in Fig. 5, the threshold on a real 60M-parameter Llama model. During training, we monitor the ratio $\|\nabla L\|/(\sigma_q \sqrt{d})$, and switch to higher-precision gradients at the 1000th iteration. When this ratio crosses $\sqrt{3}$, the loss gap to the full-precision baseline closes immediately, validating the predictive power of the threshold.

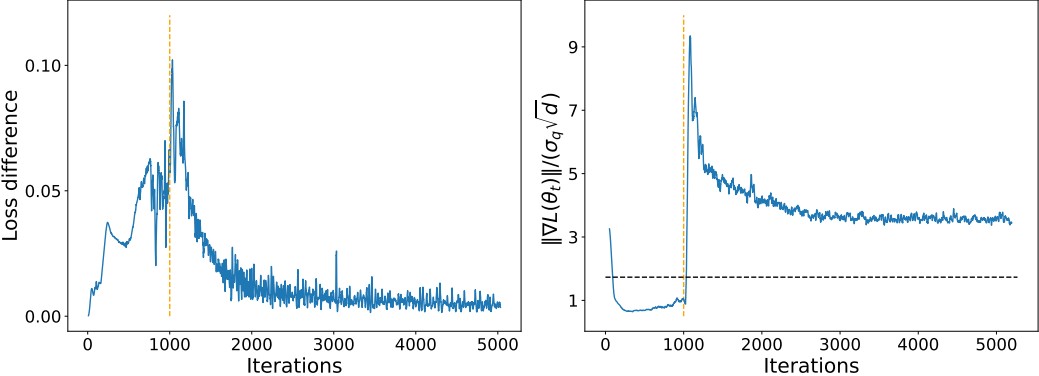

Figure 5: **Validation theoretical prediction in a Llama 60M model. (Left):** The difference between the loss curve of the baseline and a model with increasing precision mid-training (1000th iteration, vertical dashed orange line). After increasing the precision, the loss difference is completely reduced. **(Right):** Gradient-to-noise ratio with the $\sqrt{3}$ threshold (black dashed line).

## 5 Experiments

**Setup.** We used the Llama2 model [18] as our baseline. This model is a decoder-only Transformer [2] with pre-normalization RMSNorm [23], Smooth-SwiGLU activation function [8], and rotary positional embeddings [17]. We trained the models on the open-source Red Pajama dataset [5] for $1T$ tokens, maintaining hyperparameters consistent with [18], including train-test split and initialization. Specifically, we used AdamW optimizer with $\beta_1 = 0.9$, $\beta_2 = 0.95$. We used cosine learning rate schedule, with 2000 steps of warmup, peak learning rate of $3 \times 10^{-4}$ and decay to 0.1 of the peak

learning rate. We used a global batch-size of 4M tokens. All training was conducted on 256 Intel Gaudi2 devices, during $\sim 30$ days.

**FP4 training.** In Fig. 6a we present our main experiment and present the training loss of Llama2 7B with the proposed FP4 scheme, which includes the use of the NVFP4 format (block size 16, scale format E4M3) and applying SR in the neural gradients (update + backward GEMMs) and activations (update GEMM), while applying RtN for the weights (forward + backward GEMMs) and activations (forward GEMM). In Table 2 we compare the quantization settings of our work with two previous FP4 training works [19, 21], showing we are the first work that allows the acceleration of all matrix multiplication during training. In the Appendix Table 4 we show the similarity of the FP4 training losses at different seeds, showing the noise robustness of the proposed method.

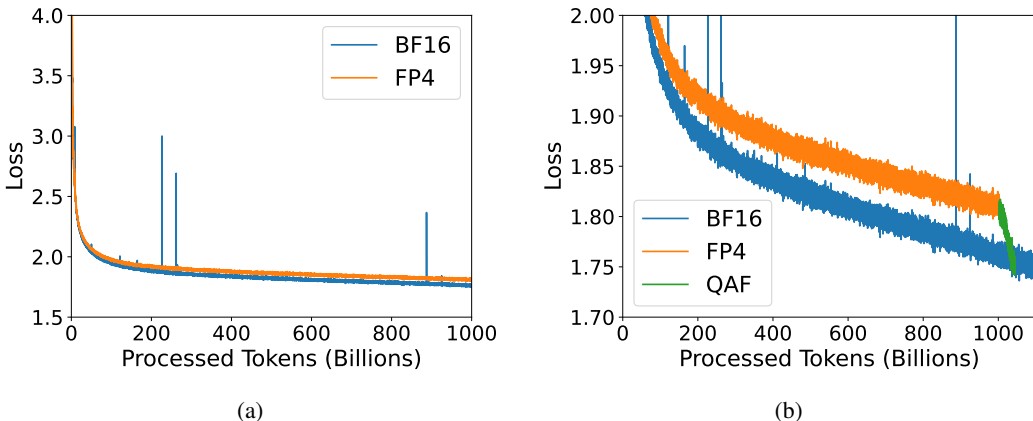

(a)                     (b)

Figure 6: **Full FP4 training a 7B model achieved same loss as BF16 baseline. (a):** Training loss of Llama2 7B using the proposed FP4 scheme which include NVFP4 format (block size 16, scale format E4M3) with SR in the neural gradients (update + backward GEMMs) and activations (update GEMM). Notice a small gap in training loss is observed. (Table 3). **(b):** Quanization aware finetuning (QAF) training loss of Llama2 7B using NVFP4 format in the forward GEMM and BF16 in the backward and update GEMMs. Notice this short QAF is able to completely close the gap with BF16 baseline.

Table 2: Comparison of the different FP4 training works. DGE refers to Differentiable Gradient Estimator, OCC to Outlier Clamp Compensation, SR to Stochastic Rounding and RHT to Random Hadamard Transform. While previous works quantize only part of the GEMMs to FP4 with additional overhead of residual sparse matrix (OCC [21]) or Hadamard transform (RHT [19]), our work is the first work that show quantization to FP4 of all GEMMs operands, without adding significant overhead.

|  | **Weight** | **Activation** | **Neural gradients** | **Tokens** |
|---|---|---|---|---|
| **[21]** | FP4+DGE+OCC | FP4+DGE+OCC | BF16 | 100B |
| **[19]** | BF16 | BF16 | MXFP4+RHT+SR | 21B |
| **Ours** | NVFP4 (RtN) | NVFP4 (RtN / SR) | NVFP4 (SR) | 1T |

**Quantization Aware Fine-tuning (QAF).** FP4 training results (Fig. 6a) in a small gap in training loss compared to the BF16 baseline. As shown in Fig. 5, increasing the precision raises the gradient-to-noise ratio higher above the critical threshold. To close the remaining gap, we introduce a brief quantization-aware finetuning (QAF) phase, where pretraining continues on the same dataset with the forward pass kept in FP4, while the backward pass is executed in BF16. Keeping the forward path in FP4 ensures that the model remains fully compatible with low-precision inference, without requiring any additional processes such as post-training-quantization. During this stage, we reset the learning rate and apply a short warmup (40 iterations) followed by a cosine decay schedule with an initial peak learning rate. In Fig. 6b we present that this short QAF can completely close the gap with BF16 baseline, achieving an average bits of 4.3 bits for GEMMs across training + QAF. In the Appendix

Table 5 we show an ablation study of the QAF length ratio required to get similar final loss as BF16. We find it decreases for larger datasets.

**Zero-shot Performance.**  Table 3 compares the zero-shot performance (accuracy and perplexity) on downstream tasks between the BF16 baseline and our FP4 model — both after the FP4 training phase (1T tokens) and after QAF phase (+40B tokens). Notice that while a small gap is observed in part of the tasks after the FP4 training phase, it is completely closed after the QAF.

Table 3: Zero shot accuracy and perplexity comparison between the BF16 baseline and the proposed FP4, both after the FP4 training phase (1T) and after QAF phase (+40B). HS refers to HellaSwag, WG refers to Winogrande, BQ refers to BoolQ, AC refers to Arc-C, PQ refers to PiQA, LA refers to Lambada, GQ refers to GPQA. IE refers to If Eval. MB refers to MBPP, TQ refers to TrivialQA. XS refers to XSum. Notice that after the QAF, the proposed FP4 model achieved on-par results with the BF16 baseline.

| Precision | Data | Accuracy ↑ | | | | | | | | | | | | Perplexity ↓ | |
| | | LA | HS | WG | AC | BQ | PQ | GQ | IE | MB | TQ | XS | Avg | Wiki | LA |
|---|---|---|---|---|---|---|---|---|---|---|---|---|---|---|---|
| BF16 | 1T | 61.52 | 68.71 | 66.54 | 38.14 | 69.33 | 76.33 | 24.54 | 33.81 | 8.2 | 34.99 | 11.97 | 45.63 | 5.54 | 6.1 |
| BF16 | 1.04T | 61.46 | 68.57 | 64.48 | 38.91 | 70.09 | 75.41 | 24.73 | 30.94 | 8.6 | 34.19 | 12.33 | 45.13 | 5.54 | 6 |
| FP4 | 1T | 58.39 | 67.31 | 64.01 | 38.65 | 69.33 | 74.86 | 24.73 | 32.13 | 6.4 | 34.3 | 12.09 | 44.46 | 5.83 | 6.77 |
| + QAF | +40B | 67.71 | 68.53 | 65.98 | 39.25 | 68.9 | 76.01 | 27.29 | 32.25 | 9.4 | 38.31 | 12.11 | 45.75 | 5.56 | 5.97 |

# 6  Discussion

This work presents the first demonstration of fully quantized FP4 training—covering weights, activations, and gradients—at large scale. Our experiments on Llama2 7B model show that, while a small gap in training loss initially appears compared to BF16, this gap can be fully closed with a short QAF phase, where the forward pass remains in FP4 and only the backward pass switches to BF16. Importantly, downstream task performance remains on par with BF16, confirming FP4's practical viability.

A key contribution is our investigation of FP4 format design. We find that NVFP4 (E4M3 with block size 16) offers the best trade-off between dynamic range and precision. Other blocks size or alternative exponent/mantissa configurations lead to instability or diminishing returns, aligning with NVIDIA Blackwell's hardware decisions.

We also introduce a split rounding strategy, using stochastic rounding only in the backward pass, which substantially improves training stability. Furthermore, our theoretical analysis identifies a critical transition point: when the full-precision gradient standard deviation falls approximately below $\sqrt{3}$ times the quantization noise, training stagnates. This insight guides the design of the final fine-tuning phase to boost the signal-to-noise ratio and match BF16 convergence.

**Limitations.**  A key limitation of this work is the lack of dedicated FP4 support in current Gaudi hardware, which prevents us from directly measuring the potential speedup and energy efficiency benefits of native FP4 execution. As a result, all experiments are conducted using FP4 simulations in Gaudi2, which incur additional overhead from precision casting and lead to longer runtimes. Based on previous FP8 works [13, 8] we expect in a rough estimation to $\sim 35 - 40\%$ time-to-train acceleration in comparison to FP8, which corresponds to $\sim 85\%$ time-to-train acceleration in comparison to the BF16 baseline. Our work centers on the LLaMA architecture, one of the most widely adopted frameworks in modern LLMs. Preliminary experiments indicate that the approach can be directly applied to Mixture-of-Experts (MoE) architectures. Further analysis of MoE models and extensions to vision tasks are reserved for future work.

## Acknowledgements

The research of DS was Funded by the European Union (ERC, A-B-C-Deep, 101039436). Views and opinions expressed are however those of the author only and do not necessarily reflect those of the European Union or the European Research Council Executive Agency (ERCEA). Neither the European Union nor the granting authority can be held responsible for them. DS also acknowledges the support of the Schmidt Career Advancement Chair in AI.

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

.


# Appendix

## A  Broader impacts

Accelerating the runtime of large language models (LLMs) plays a pivotal role in shaping modern digital experiences, especially as systems like ChatGPT and Gemini become increasingly embedded in everyday applications. Improving their speed and efficiency addresses a major bottleneck in large-scale AI deployment. In addition to boosting performance and reducing memory demands, faster models broaden accessibility, enabling more users to adapt and innovate with LLMs for their specific needs. However, this increased accessibility also raises concerns about potential misuse, underscoring the need for responsible development and oversight.

## B  Analysis of quantized SGD

### B.1  Analysis of quantized SGD with stochastic rounding

We study how quantization noise affects the expected loss decrease during gradient descent.

Let $L(\theta)$ be a twice-differentiable scalar loss function on $\mathbb{R}^d$.

**Step 1: Start with the Taylor expansion of the loss.**  We consider a small step $\Delta\theta$ from $\theta_t$. The second-order Taylor expansion of $L$ at $\theta_t$ is:

$$L(\theta_t + \Delta\theta) = L(\theta_t) + \nabla L(\theta_t)^T \Delta\theta + \frac{1}{2}\Delta\theta^T H(\theta_t)\Delta\theta + \cdots$$

where $H(\theta_t) = \nabla^2 L(\theta_t)$.

**Step 2: Apply a quantized gradient update.**  We use a noisy estimate of the gradient due to quantization:

$$g_q = \nabla L(\theta_t) + \varepsilon,$$

where $\varepsilon$ is quantization noise. The update rule becomes:

$$\theta_{t+1} = \theta_t - \eta\, g_q \quad \Rightarrow \quad \Delta\theta = -\eta\, g_q.$$

**Step 3: Substitute into the Taylor expansion.**  Plugging $\Delta\theta = -\eta\, g_q$ gives

$$L(\theta_{t+1}) \approx L(\theta_t) - \eta\, \nabla L(\theta_t)^T g_q + \frac{1}{2}\eta^2\, g_q^T H(\theta_t)\, g_q.$$

**Step 4: Expectation over quantization noise.**  Under stochastic rounding, the noise $\varepsilon$ satisfies

$$\mathbb{E}[\varepsilon] = 0, \qquad \mathbb{E}[\varepsilon\varepsilon^T] = \sigma_q^2 I.$$

Taking expectations in the Taylor expansion gives

$$\mathbb{E}[L(\theta_{t+1})] \approx L(\theta_t) - \eta\, \nabla L(\theta_t)^T \mathbb{E}[g_q] + \tfrac{1}{2}\eta^2\, \mathbb{E}\big[g_q^T H(\theta_t)\, g_q\big].$$

Since

$$\mathbb{E}[g_q] = \mathbb{E}[\nabla L(\theta_t) + \varepsilon] = \nabla L(\theta_t),$$

the linear term simplifies to

$$-\eta\, \nabla L(\theta_t)^T \nabla L(\theta_t) = -\eta\, \|\nabla L(\theta_t)\|^2.$$

Next,

$$\mathbb{E}[g_q g_q^T] = \mathbb{E}\big[(\nabla L + \varepsilon)(\nabla L + \varepsilon)^T\big] = \nabla L\, \nabla L^T + \sigma_q^2 I,$$

so

$$\mathbb{E}[g_q^T H(\theta_t)\, g_q] = \mathrm{tr}\big(H(\theta_t)\, \mathbb{E}[g_q g_q^T]\big) = \nabla L(\theta_t)^T H(\theta_t)\, \nabla L(\theta_t) + \sigma_q^2\, \mathrm{tr}\big(H(\theta_t)\big).$$

Putting everything together,

$$\mathbb{E}[L(\theta_{t+1})] = L(\theta_t) - \eta\, \|\nabla L(\theta_t)\|^2 + \tfrac{1}{2}\eta^2 \Big(\nabla L(\theta_t)^T H(\theta_t)\, \nabla L(\theta_t) + \sigma_q^2\, \mathrm{tr}\big(H(\theta_t)\big)\Big).$$

**Step 5: Convergence dynamics with SR**  From Step 4, the expected change in loss is:

$$\mathbb{E}[L(\theta_{t+1}) - L(\theta_t)] \approx -\eta\|\nabla L(\theta_t)\|_2^2 + \tfrac{1}{2}\eta^2\left(\nabla L(\theta_t)^T H(\theta_t)\nabla L(\theta_t) + \sigma_q^2 \mathrm{tr}(H(\theta_t))\right).$$

This can be written as:

$$\mathbb{E}[L(\theta_{t+1}) - L(\theta_t)] \approx \underbrace{-\left(\eta\|\nabla L(\theta_t)\|_2^2 - \tfrac{1}{2}\eta^2\nabla L(\theta_t)^T H(\theta_t)\nabla L(\theta_t)\right)}_{\text{useful descent component}} + \underbrace{\tfrac{1}{2}\eta^2\sigma_q^2 \mathrm{tr}(H(\theta_t))}_{\text{quantization noise effect}}.$$

The useful descent component is negative if:

$$\eta\|\nabla L(\theta_t)\|_2^2 > \tfrac{1}{2}\eta^2\nabla L(\theta_t)^T H(\theta_t)\nabla L(\theta_t) \Rightarrow \eta < \frac{2\|\nabla L(\theta_t)\|_2^2}{\nabla L(\theta_t)^T H(\theta_t)\nabla L(\theta_t)}.$$

A more conservative condition is:

$$\eta < \frac{2}{\lambda_{\max}(H(\theta_t))}.$$

**Step 6: Optimal step size $\eta^*$.**  To find the optimal step size, we define

$$U(\eta) = \mathbb{E}[L(\theta_{t+1}) - L(\theta_t)] = -\eta\|\nabla L(\theta_t)\|_2^2 + \tfrac{1}{2}\eta^2\left(\nabla L(\theta_t)^T H(\theta_t)\nabla L(\theta_t) + \sigma_q^2 \mathrm{tr}(H(\theta_t))\right).$$

Setting its derivative to 0:

$$\frac{dU}{d\eta} = -\|\nabla L(\theta_t)\|_2^2 + \eta\left(\nabla L(\theta_t)^T H(\theta_t)\nabla L(\theta_t) + \sigma_q^2 \mathrm{tr}(H(\theta_t))\right) = 0.$$

Solving for $\eta^*$:

$$\eta^* = \frac{\|\nabla L(\theta_t)\|_2^2}{\nabla L(\theta_t)^T H(\theta_t)\nabla L(\theta_t) + \sigma_q^2 \mathrm{tr}(H(\theta_t))}.$$

**Step 7: Training with optimal step size $\eta^*$.**  Substitute $\eta^* = \frac{A}{B}$ back into

$$U(\eta) = -\eta\,A + \tfrac{1}{2}\,\eta^2\,B,$$

where

$$A = \|\nabla L(\theta_t)\|_2^2, \qquad B = \nabla L(\theta_t)^T H(\theta_t)\,\nabla L(\theta_t) + \sigma_q^2\,\mathrm{tr}\big(H(\theta_t)\big).$$

Then

$$U(\eta^*) = -\frac{A}{B}\,A + \tfrac{1}{2}\Big(\frac{A}{B}\Big)^2 B = -\frac{A^2}{B} + \frac{1}{2}\frac{A^2}{B} = -\frac{1}{2}\frac{A^2}{B},$$

i.e.

$$U(\eta^*) = -\frac{\|\nabla L(\theta_t)\|_2^4}{2\big(\nabla L(\theta_t)^T H(\theta_t)\,\nabla L(\theta_t) + \sigma_q^2\,\mathrm{tr}(H(\theta_t))\big)}.$$

Let $X = \nabla L^T H \nabla L$, $Y = \mathrm{tr}(H)$ and $Z = \|\nabla L\|_2^4$. Then

$$U(\eta^*) = -\frac{Z}{2\big(X + \sigma_q^2 Y\big)}.$$

**Step 8: Maximum sensitivity to noise.**  We start with the sensitivity

$$f(\sigma_q) = \frac{\partial U(\eta^*)}{\partial \sigma_q} = \frac{Z\,Y\,\sigma_q}{\big(X + Y\,\sigma_q^2\big)^2}.$$

To find its maximum, compute the derivative w.r.t. $\sigma_q$:

$$\frac{df}{d\sigma_q} = Z\,Y\,\frac{(X + Y\sigma_q^2)^2 - \sigma_q \cdot 2\,(X + Y\sigma_q^2)\cdot(2Y\sigma_q)}{(X + Y\sigma_q^2)^4}.$$

We factor this expression and simplify it:

$$\frac{df}{d\sigma_q} = \frac{Z\,Y\,(X + Y\sigma_q^2)\big[(X + Y\sigma_q^2) - 4Y\sigma_q^2\big]}{(X + Y\sigma_q^2)^4} = \frac{Z\,Y\,[\,X - 3Y\sigma_q^2\,]}{(X + Y\sigma_q^2)^3}.$$

Set this to zero to locate the minimum:

$$X - 3Y\sigma_q^2 = 0 \implies \sigma_q^2 = \frac{X}{3Y}.$$

Thus the critical noise level is

$$\sigma_{\text{critical}}^2 = \frac{X}{3Y}.$$

That is:

$$\sigma_{\text{critical}}^2 = \frac{\nabla L(\theta_t)^T H(\theta_t) \nabla L(\theta_t)}{3\,\text{tr}\big(H(\theta_t)\big)}.$$

Finally, assuming[3]

$$\frac{\nabla L(\theta_t)^T H(\theta_t) \nabla L(\theta_t)}{\|\nabla L(\theta_t)\|_2^2} \approx \frac{\text{tr}\big(H(\theta_t)\big)}{d},$$

we get

$$\|\nabla L(\theta_t)\|_2^2 \approx 3\,d\,\sigma_{\text{critical}}^2 \implies \|\nabla L(\theta_t)\|_2 \approx \sqrt{3d}\,\sigma_{\text{critical}}.$$

Therefore,

$$\sigma_{\text{critical}} = \frac{\|\nabla L(\theta_t)\|_2}{\sqrt{3d}}.$$

In other words, once the average per-coordinate gradient falls to $\sqrt{3}$ times the quantization-noise std, FP4 gradients lose efficacy and it is time to switch to higher precision. As shown in Figure 4, setting the noise std to $k \cdot \sigma_{\text{critical}}$ with $k = 2.0, 1.0, 0.5$ yields markedly different convergence behaviors around that threshold. In Appendix B.2 we show a similar analysis without SR, which shows that the "useful descent" vanishes to zero as we train, while the biased-noise term remains even after long training..

## B.2 Impact of nonzero mean noise (deterministic rounding)

In this section, we exemplify the problem with biased quantization schemes (such as RtN), in a simple scalar optimization problem with a quadratic loss

$$L(\theta) = \tfrac{1}{2}\lambda(\theta - \theta^*)^2, \implies \nabla L(\theta) = \lambda(\theta - \theta^*),$$

and a step size update with quantization noise $\varepsilon$ whose mean is $\mu_\varepsilon = \mathbb{E}[\varepsilon] \neq 0$:

$$\theta_{t+1} = \theta_t - \eta\big(\nabla L(\theta_t) + \varepsilon\big) \implies \mathbb{E}[\theta_{t+1}] = \mathbb{E}[\theta_t] - \eta\big(\lambda(\mathbb{E}[\theta_t] - \theta^*) + \mu_\varepsilon\big).$$

Define the error

$$e_t \triangleq \mathbb{E}[\theta_t] - \theta^*.$$

Then

$$e_{t+1} = \mathbb{E}[\theta_{t+1}] - \theta^* = \big[\mathbb{E}[\theta_t] - \eta(\lambda e_t + \mu_\varepsilon)\big] - \theta^* = e_t - \eta\lambda e_t - \eta\mu_\varepsilon.$$

Therefore,

$$e_{t+1} = (1 - \eta\lambda)\,e_t - \eta\,\mu_\varepsilon.$$

Unrolling this recursion gives, for $a = 1 - \eta\lambda$,

$$e_n = a^n e_0 - \eta\,\mu_\varepsilon \sum_{k=0}^{n-1} a^k.$$

Since $\sum_{k=0}^{n-1} a^k = \frac{1-a^n}{1-a}$ and $1 - a = \eta\lambda$, we get

$$e_n = a^n e_0 - \frac{\eta\,\mu_\varepsilon}{\eta\lambda}\big(1 - a^n\big) = a^n e_0 - \frac{\mu_\varepsilon}{\lambda}\big(1 - a^n\big).$$

---

[3]In the high-dimensional regime ($d, N \to \infty$ and $d/N \to \lambda$, where $N$ is number of training samples), previous works [14] showed, using random-matrix theory and empirical results, that the bulk of the Hessian values can be approximately represented by the Marchenko–Pastur distribution, especially for small loss values. Thus, when $\lambda \ll 1$ (which is the common regime for LLMs), this Marchenko–Pastur distribution implies that the bulk of Hessian eigenvalues concentrates near their mean $\text{tr}(H)/d$. Empirical investigations [16, 10] confirm that the gradient predominantly occupies this bulk subspace rather than aligning with the few extreme modes. Hence the curvature experienced in the gradient direction approximates the average eigenvalue.

The loss at step $n$ is

$$L_n = L(\mathbb{E}[\theta_n]) = \tfrac{1}{2}\lambda e_n^2 = \frac{\lambda}{2}\left(a^n e_0 - \frac{\mu_\varepsilon}{\lambda}(1 - a^n)\right)^2.$$

As $n \to \infty$, $a^n \to 0$ (for $a < 0$, which is required for successful optimization), yielding the stationary error and residual loss

$$e_\infty = -\frac{\mu_\varepsilon}{\lambda}, \qquad L_\infty = \frac{\mu_\varepsilon^2}{2\lambda}.$$

Thus, instead of converging to $\theta^*$ with zero loss, biased SGD settles at

$$\mathbb{E}[\theta_\infty] = \theta^* - \frac{\mu_\varepsilon}{\lambda},$$

and leaves a residual loss

$$L(\mathbb{E}[\theta_\infty]) = \frac{\mu_\varepsilon^2}{2\lambda}.$$

## C  Additional Experimental Results

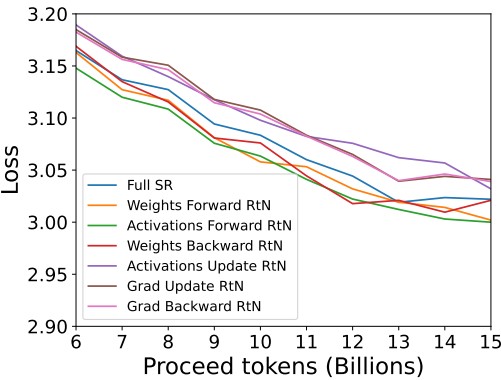

Figure 7: **Comparison of different rounding schemes when training a 350M Llama model using NVFP4 format.** In each graph, we apply RtN in one of the six elements in one of the GEMMs while the rest use SR. Notice that applying RtN to neural gradients during both the 'Update' and 'Backward' GEMMs, and to the activations during the 'Update' GEMM leads to higher training loss, while applying RtN to the other components has the opposite effect, reducing the loss.

Table 4: Training loss of Llama 125M over 30B tokens with different seeds, yield a similar loss, resulting in a standard deviation of 0.001.

| Training loss | Seed |
|---|---|
| 3.03 | 1337 |
| 3.027 | 1234 |
| 3.025 | 2345 |
| 3.027 | 3456 |
| 3.029 | 4567 |

Table 5: Ablation study to determine how many tokens are required to reach the same loss as the BF16 baseline. Notice the QAF ratio decrease for larger dataset. In all QAF experiments we use the peak learning rate equal to the last learning rate in the FP4 training.

| Starting point | QAF length | Ratio |
|---|---|---|
| 200B | 20B | 10% |
| 500B | 28B | 5.6% |
| 1T | 40B | 4% |

