# OpenReview forum: "FP4 All the Way: Fully Quantized Training of Large Language Models"
_NeurIPS.cc/2025/Conference — NeurIPS 2025 spotlight_

### Official Review · Reviewer_vv7s · 2025-07-02

**Clarity:** 3
**Significance:** 4
**Originality:** 3
**Rating:** 5
**Confidence:** 4

**Summary:**

The authors in this work demonstrate end-to-end 4-bit FP training at LLM scale (7B, 200B tokens), extending FQT capabilities beyond existing FP8 approaches. The authors claim their work to be the first line of works demonstrate complete FP4 end to end training. The authors evaluate key design choices (block size, scaling format, rounding modes) and identify NVFP4 (E2M1 data with shared E4M3 scale in blocks of 16) with mixed rounding (stochastic for backward/update, round-to-nearest for forward) as optimal. Lastly, they successfully train a 7B-parameter model on Intel Gaudi2 accelerators to performance comparable to BF16.

**Questions:**

1. Please refer to weakness points (1), (2), (3). Including ablations to address these concerns would strengthen their claims.
2. How does the FP4-trained model compare to FP16 baseline on downstream tasks (e.g., QA, summarization, code generation)? Does full quantization affect generalization across domains or modalities? The authors are encouraged to include a few difficult baselines to demonstrate the at-par performance wrt BF16 baseline.
3. How sensitive is the training stability to the √3 gradient-to-noise threshold? Also, can the authors elaborate on how and when precision switching is triggered in practice?
4. How does the method behave under varying batch sizes? Smaller batches often lead to noisier gradients, does this make FP4 more prone to training collapse? Also, how sensitive is the FP4 training pipeline to initialization schemes?
5. Given that the authors use of stochastic rounding for backward pass, how reproducible are the training runs across seeds or hardware environments? Are there significant variance in convergence / final performance for different random seeds?

**Ethical Concerns:**

["NO or VERY MINOR ethics concerns only"]

**Final Justification:**

The authors answered most of my concerns with respect to understanding the sensitivity of the algorithm to different batch sizes, seeds and architectures. The proposed method seems promising however understanding and discussing its usecase and limitations is important for generalization.

**Limitations:**

Yes, the authors have discussed limitations of their method.

As a suggestion, the authors are encouraged to further consider robustness related limitations, especially the potential vulnerability of their approach to noisy gradients or its applicability in vision and multimodal settings, where different modalities may exhibit different gradient distributions or magnitudes that could affect the stability of FP4 training.

**Quality:**

3

**Strengths And Weaknesses:**

This is a timely work which focuses on lower precision training of LLMs. If generalizable across models, it could have moderate to high impact on an industrial scale.

Strengths:
1. This authors claim their work to be the first to enable end-to-end 4-bit FP training at LLM scale (7B, 200B tokens), extending FQT capabilities beyond existing FP8 approaches.
2. The authors systematically explore block sizes (8 to 128), scaling formats (MXFP4 vs NVFP4), and rounding schemes, concluding NVFP4 with block size 16 delivers optimal stability. Also, they identify that rounding strategy using stochastic rounding for gradients and round-to-nearest for forward passes, significantly improves stability, especially for backward and update paths. These insights can be very beneficial for the community to further push ultra low precision training of large scale models.
3. The paper introduces a theoretically derived and empirically validated threshold for gradient norms droping below √3 × quantization noise, suggesting dynamic precision adaptation.
4. Demonstrated training of a 7B LLM on Gaudi2 chips with only FP4 showing practical viability. The trained model also achieved downstream results on par with BF16 baselines.

Weakness:
1. Although the 7B training shown performs at par with BF16 baseline,  it's unclear if FP4 FQT scales to larger models (30B+) or more diverse architectures beyond the 200B token setting. Additionally, as recent work [1] has also proposed an end-to-end FP4 training framework, the authors are encouraged to clarify the distinctions and provide insights into how their approach compares or complements this effort.
2. Due to the use of shared scaling across blocks, stochastic rounding, and dynamic precision switching, this could likely incurs non-trivial computational cost. Providing a quantitative analysis of these costs would offer clearer insight into the trade-offs between efficiency and precision stability.
3. The authors are also encouraged to include comparisons to all-nearest or all-stochastic rounding configurations. This would be beneficial in understanding the relative comparison of different schemes for stable training.

[1] Wang, Ruizhe, et al. "Optimizing Large Language Model Training Using FP4 Quantization." arXiv preprint arXiv:2501.17116 (2025).

---

> ### Author Rebuttal · Authors · 2025-07-31
>
> $\textbf{Q1a}:$ Although the 7B training shown performs at par with BF16 baseline, it's unclear if FP4 FQT scales to larger models (30B+) or more diverse architectures beyond the 200B token setting.
>
> $\textbf{A1a}:$ We continued the training presented in Fig. 6 to 1T tokens, followed by 40B tokens of QAF. We achieved training loss of 1.752 (FP4) Vs 1.76 (BF16), showing the ability of the proposed method to work in a larger scale dataset. Then we compare the BF16 baseline with FP4 model on downstream tasks (similar to Table 3), including new tasks as required by the reviewer - GPQA (GQ), IfEval (IF), MBPP (MB), TrivialQA (TQ), XSum (XS):
>
> | Precision | Tokens | LA    | HS    | WG    | AC    | BQ    | PQ    | GQ    | IE    | MB   | TQ   | XS   |  *   | Avg   |  *   | Wiki(PPL) | LA(PPL) |
> |-----------|--------|-------|-------|-------|-------|-------|-------|-------|-------|------|------|------|-----|--------|-----|------------|----------|
> | BF16      | 1T     | 61.52 | 68.71 | 66.54 | 38.14 | 69.33 | 76.33 | 24.54 | 33.81 | 8.2  | 34.99| 11.97| *   | 45.63 |  *   | 5.54       | 6.10     |
> | BF16      | 1.04T  | 61.46 | 68.57 | 64.48 | 38.91 | 70.09 | 75.41 | 24.73 | 30.94 | 8.6  | 34.91| 12.33|  *   | 45.13 |   *  | 5.54       | 6.00     |
> | FP4       | 1T     | 58.39 | 67.31 | 64.01 | 38.65 | 69.33 | 74.86 | 24.73 | 32.13 | 6.4  | 34.30| 12.09|   *  | 44.46 |  *   | 5.83       | 6.77     |
> | +QAF      | +40B   | 61.71 | 68.53 | 65.98 | 39.25 | 68.90 | 76.01 | 27.29 | 32.25 | 9.4  | 38.31|  12.11   |  *   | 45.75 |  *   | 5.56       | 5.97     |
>
> We agree with the reviewer that it will be interesting to see the proposed method on larger models (30B+), however it requires more compute resources than we currently have: just our experiments on a 7B model for 1T tokens took around 30 days on 256 devices.
>
>
>
> $\textbf{Q1b}:$ Additionally, as recent work [1] has also proposed an end-to-end FP4 training framework, the authors are encouraged to clarify the distinctions and provide insights into how their approach compares or complements this effort.
>
> $\textbf{A1b}:$ Please note we already cited this work as [18], and its distinctions from us are explained in lines 103-109 in the "related work" section and Table 2.  Specifically, our work is the first work that shows full fp4 training, i.e, quantization to fp4 of the 3 general-matrix-multiplications (GEMMs) (forward, backward, and update - Equations 1,2,3). In contrast, [18] quantized only the forward GEMM, i.e can accelerate only $\sim 33$% of the GEMMs, which dramatically reduces the potential acceleration. Moreover, they trained only up to 100B tokens, which is an under-trained regime; we extend it to a 10x larger dataset. Please see Table 2 for a full comparison with [18].
>
> $\textbf{Q2}:$ Due to the use of shared scaling across blocks, stochastic rounding, and dynamic precision switching, this could likely incurs non-trivial computational cost. Providing a quantitative analysis of these costs would offer clearer insight into the trade-offs between efficiency and precision stability.
>
> $\textbf{A2}:$ Shared scaling across blocks is supported natively in hardware in Blackwell architecture without any acceleration cost, i.e it allows a 2x acceleration in comparison to FP8 and 4x in comparison with BF16. We expect additional AI accelerators also to support it natively without additional cost.
>
> Stochastic rounding is also supported natively in the latest Nvidia's CUBLASS (v8.8) or in Gaudi2, without any significant overhead.
>
> Notice we do not use dynamic precision switching. We only increase once the precision of the backward (Eq.2) and update (Eq.3) GEMMs when starting the QAF process. In order to calculate the total cost, we can calculate the average bits operations during all process (pretraining + QAF), as we detail next.
> Recall we extended the pretraining phase to 1T tokens, followed by a 40B-token QAF phase. During pretraining all three GEMMs operate in FP4, while during QAF only one of the three GEMMs uses FP4 (with the remaining two in BF16). The resulting average-bits calculation is:$ [(1000\times3\times4) + (40\times1\times4) + (40\times2\times16)] / [1040\times3] = 4.3$ bits. We will include this result in the camera-ready version.
>
>
>
> $\textbf{Q3}:$ The authors are also encouraged to include comparisons to all-nearest or all-stochastic rounding configurations. This would be beneficial in understanding the relative comparison of different schemes for stable training.
>
> $\textbf{A3}:$ In Fig. 3 we show the comparison to all-nearest (marked as "full RtN"). Moreover, we conducted an additional experiment similar to Fig. 3, where we independently applied RtN to one of the six elements while the remaining elements used SR. The resulting losses are as follows: Full SR: 3.022; Weights Forward RtN: 3.002 ; Activations Forward RtN: 3 ; Weights Backward RtN: 3.021 ; Activations Backward RtN: 3.032 ; Gradient Update RtN: 3.041 ; Gradient Backward RtN: 3.039.
>
> This experiment highlights the effectiveness of the proposed regime, as described in Equations 4, 5, and 6, where SR is applied only to neural gradients during the update and backward GEMMs, and to activations during the update GEMM. We will include this experiment in the camera-ready version.
>
> $\textbf{Q4}:$ How does the FP4-trained model compare to FP16 baseline on downstream tasks (e.g., QA, summarization, code generation)? Does full quantization affect generalization across domains or modalities? The authors are encouraged to include a few difficult baselines to demonstrate the at-par performance wrt BF16 baseline.
>
> $\textbf{A4}:$ Please see the response A1a, where we show the evaluation on additional tasks for summarization (xsum), QA (trivial qa) and code generation (mbpp) as required by the reviewer.
>
>
>
> $\textbf{Q5}:$ How sensitive is the training stability to the $\sqrt{3}$ gradient-to-noise threshold? Also, can the authors elaborate on how and when precision switching is triggered in practice?
>
> $\textbf{A5}:$ The $\sqrt{3}$ gradient-to-noise threshold serves as a useful indicator for assessing the effectiveness of quantized training and identifying when it is no longer beneficial to continue training in low precision. Figure 5 illustrates a scenario where this threshold is crossed, suggesting that full low-precision training should be stopped. In contrast, for the experiment shown in Figure 6, the gradient-to-noise ratio remains above the threshold. This was a significant positive result that justified our decision to allocate the resources to continue training in 4-bit all the way to 1T tokens. It also demonstrates that with the optimal format (NVFP4) and our split rounding strategy, large-scale FP4 training can be successfully completed without even encountering the stagnation point that our theory predicts.
>
>
> $\textbf{Q6}:$ How does the method behave under varying batch sizes? Smaller batches often lead to noisier gradients, does this make FP4 more prone to training collapse? Also, how sensitive is the FP4 training pipeline to initialization schemes?
>
> $\textbf{A6}:$ Following the reviewer’s request, we run 3 runs with different micro-batch sizes (1K,10K,20K) in a 125M model trained over 30B tokens using the proposed scheme on Gaudi2 devices, we get similar training losses: 3.027, 3.025, 3.03
> We believe that the use of per-block quantization (in contrast to per-tensor quantization) reduces the effect of batch size, mentioned by the reviewer.
> Regarding different initializations, please refer to response A7 below, where we tested multiple seeds (including different initializations) and did not observe any convergence issues.
>
> $\textbf{Q7}:$ Given that the authors use of stochastic rounding for backward pass, how reproducible are the training runs across seeds or hardware environments? Are there significant variance in convergence / final performance for different random seeds?
>
> $\textbf{A7}:$ Following the reviewer’s request, we conducted 5 runs with different seeds on a 125M model trained over 30B tokens using the proposed scheme on Gaudi2 devices. The resulting losses are: 3.03, 3.027, 3.025, 3.027,3.029 yield a loss standard deviation of 0.001, which is not significant. Moreover, we ran the same experiment on an A100 GPU and obtained an equivalent loss of 3.024. As the reviewer can see, we did not observe any convergence issues across different seeds or hardware platforms. We plan to include these ablation studies in the camera-ready version.
>
> $\textbf{Q8}:$ As a suggestion, the authors are encouraged to further consider robustness related limitations, especially the potential vulnerability of their approach to noisy gradients or its applicability in vision and multimodal settings, where different modalities may exhibit different gradient distributions or magnitudes that could affect the stability of FP4 training.
>
> $\textbf{A8}:$ Thank you for this suggestion. We will add it to the limitations section, explicitly emphasizing that our experiments focus on language tasks, while vision tasks are left for future work.

---

> > ### Comment · Reviewer_vv7s · 2025-08-05
> >
> > Thank you to the authors for the detailed rebuttal. It addresses most of my questions, and I appreciate the additional experiments on downstream tasks, batch size sensitivity,  and reproducibility. These additions experiments clarify many of the technical points I had.
> >
> > I have a clarifying question based on the generalizability of the training method:
> > 1. While your current work is based on the LLaMA2 architecture, do you foresee any specific challenges in extending FP4 training to other architectures such as Mixture-of-Experts (MoE) or state-space models like Mamba?
> > In particular, MoE models involve sparse activation and expert-specific gradients, while SSM-based models like Mamba may have different gradient magnitudes or temporal dependencies. From a gradient stability and quantization noise perspective, do you anticipate that these differences would affect the effectiveness or robustness of FP4 training?

---

> > > ### Author Response · Authors · 2025-08-06
> > > **Response to reviewer vv7s**
> > >
> > > Thank you for your positive feedback.
> > >
> > > Regarding your question — we’ve begun some initial experiments with MoE architectures, specifically using OLMoE. We trained a 500M parameter model (with 100M active parameters) and observed losses of 2.72 (BF16) vs. 2.732 (FP4). In the FP4 version, we quantize both the attention and the experts, but we found that the routing layer (less than 0.1% of the total parameters) is particularly sensitive to quantization, so we chose to keep it in higher precision. We plan to further explore this direction and assess the potential benefits of the model’s sparse activation.
> > >
> > > As for Mamba — this is indeed an interesting direction, but one we haven’t yet explored. We’ll consider it for future work.
> > >
> > > We hope this addresses your concerns. If so, we’d greatly appreciate it if you would consider increasing your rating.

---

### Official Review · Reviewer_TyRD · 2025-07-02

**Clarity:** 3
**Significance:** 4
**Originality:** 3
**Rating:** 5
**Confidence:** 4

**Summary:**

This paper demonstrates the first fully quantized training (FQT) of large language models using only FP4 precision for all matrix operations, including weights, activations, and gradients on NVFP4 format, with mixed rounding modes, round-to-nearest in the forward pass and stochastic rounding in the backward pass, to ensure stable training. They also propose a straightforward gradient noise threshold heuristic, based on a simple quadratic loss, to determine when to switch from FP4 to higher precision during training. This is implemented as a brief mixed-precision "Quantization-Aware Fine-tuning" phase using BF16. The study features systematic ablation experiments on quantization format, block size, and rounding strategies, achieving competitive zero-shot performance on multiple downstream tasks. Although currently emulated on Gaudi2 hardware, the method is designed as a blueprint for efficient FP4 training on future hardware such as Blackwell.

**Questions:**

I would like to kindly ask the authors to address the issues stated in the “Weaknesses” section. I would gladly increase my scores if the issues are properly addressed.

**Ethical Concerns:**

["NO or VERY MINOR ethics concerns only"]

**Final Justification:**

The author's rebuttal has addressed all of my concerns, and in addition, they have provided stronger evaluations which better justify their method, and I think this paper is a high-impact paper in the field.

**Limitations:**

Yes.

**Paper Formatting Concerns:**

No major concern.

**Quality:**

3

**Strengths And Weaknesses:**

Strengths:
1. First end-to-end FP4 fully-quantized training at non-toy scale, bridging the gap between FP8 and the state-of-the-art hardware.
2. Clear exploration and ablations on scale formats, block sizes, and per-operand rounding choices.
3. A full 7B parameter pre-training run on an open-source corpus with +200B tokens, demonstrating realism beyond toy tasks
4. The noise threshold provides an actionable heuristic for when to up-switch precision, matching empirical behavior.




Weaknesses:
1. Lacking an isolated round-to-nearest impact per matrix, similar to Fig. 3. Without it, we cannot decide where SR would beat RtN.
2. While the theoretical derivation in 4.1 is intuitive and somehow hints why we might have to switch to a higher precision, it lacks proper argument on why this critical measure is the direct cause of instability, i.e. it can be concluded that the worst expected error happens at this critical variance, yet doesn’t imply that the absolute value of $E[\Delta L]$ is critically high when variance reaches this critical value. The derivation does not bound the absolute loss change $E[\Delta L]$, nor prove divergence or “instability” at that point.
3. The 220 B-token run yields a token-to-parameter ratio of ~30×, which scaling-law literature[3,4] deems under-trained and thus less likely to expose late-stage instabilities; training a smaller model on the same token budget would be a stronger stress test.
4. No measured total BOPS, throughput, or average bit-width across FQT + QAF; a quick estimate suggests roughly ~5.2 bits overall.
5. Experiments rely on Gaudi2 simulation rather than widely deployed Blackwell hardware, so real-world performance gains remain speculative.
6. Stochastic-rounding implementation is unspecified; the probability mass function is unclear.
7. Related-work survey omits two sub-FP8 methods, QuEST[1] and AdaBin[2], which should appear in Table 2.
8. Training hyperparameters are incomplete; learning rate schedules, batch sizes, and optimizer settings, among others, are missing.
9. All results are based on a single seed. Table 3 lacks error bars or confidence intervals.
10. No empirical study on how long QAF must run, nor on LR and optimiser sensitivity in FP4.
11. No exploration of combining the method with PTQ for additional compression. i.e., applying post-training quantisation methods such as SmoothQuant or GPTQ on top of the FP4-trained model, and inspect the combined effect on accuracy.




Minor text issue:
1. Possible typo on Fig.1 and Fig.2 x-axis label “proceed” -> “processed”
2. Line 238: “close completely” -> “completely close”
3. Typo: “trainng” in Fig. 6 caption.




References:

[1] QuEST: Stable Training of LLMs with 1-Bit Weights and Activations, Panferov et al.

[2] AdaBin: Improving Binary Neural Networks with Adaptive Binary Sets, Tu et al.

[3] Scaling Laws for Neural Language Models, Kaplan et al.

[4] Training Compute-Optimal Large Language Models, Hoffmann et al.

---

> ### Author Rebuttal · Authors · 2025-07-31
>
> $\textbf{General comment:}$ In response to the reviewer's questions, we would like to clarify a potential misunderstanding regarding the different training phases:
> (1) Long Pretraining: All three general matrix multiplications (GEMMs) described in Equations 1, 2, and 3 are quantized to FP4.
> (2) Short QAF: The forward GEMM (Eq.1) is quantized to FP4, while the backward (Eq.2 ) and update (Eq.3) steps are performed in BF16.
>
> This means the final model contain FP4 weights and can run inference directly in the FP4 datatype without requiring any additional processes (such as PTQ). We plan to improve and clarify this explanation in the camera-ready version.
>
> $\textbf{Q1}:$ "Lacking an isolated round-to-nearest impact per matrix, similar to Fig. 3..."
>
> $\textbf{A1}:$ In response to the reviewer’s request, we conducted an additional experiment similar to Fig. 3, where we independently applied RtN to one of the six elements while the remaining elements used SR. The resulting training losses are as follows: Full SR: 3.022; Weights Fwd RtN: 3.002 ; Activations Fwd RtN: 3 ; Weights Bwd RtN: 3.021 ; Activations Bwd RtN: 3.032 ; Gradient Update RtN: 3.041 ; Gradient Backward RtN: 3.039.
>
> This experiment highlights the effectiveness of the proposed regime, as described in Equations 4, 5, and 6, where SR is applied only to neural gradients during the update and backward GEMMs, and to activations during the update GEMM. We will include this experiment in the camera-ready version
>
> $\textbf{Q2}:$ "While the theoretical derivation in 4.1 is intuitive ..."
>
> $\textbf{A2}:$ We appreciate the reviewer’s comment and agree with their observation. Our analysis identifies a threshold for training ineffectiveness (i.e., when progress stalls) rather than proving instability. This occurs when the gradient signal becomes too weak to overcome the quantization noise, causing the optimization process to plateau.
> Because our goal is to define this practical point of failure, the derivation in Section 4.1 is intended as a local metric to guide training decisions, not as a global proof of convergence. Consequently, we do not claim to bound the absolute change in loss or present a full convergence guarantee. As shown in Figure 5, this interpretation is supported empirically: when the metric falls below the proposed threshold, continuing training in low precision is no longer effective, and an intervention such as increasing the numerical precision is required to resume progress.
>
>
> $\textbf{Q3}:$ "The 220 B-token..."
>
> $\textbf{A3}:$ We have extended the primary training of the 7B model (Figure 6) to 1 trillion tokens, followed by 40B tokens of QAF, which requires approximately 30 days on 256 devices. We achieved training loss of 1.752 (FP4) Vs 1.76 (BF16). Then we compare the BF16 baseline  with FP4 model on downstream tasks (similar to Table 3), including new tasks - GPQA (GQ), IfEval (IF), MBPP (MB), TrivialQA (TQ), XSum (XS):
>
> | Precision | Tokens | LA    | HS    | WG    | AC    | BQ    | PQ    | GQ    | IE    | MB   | TQ   | XS   |  *   | Avg   |  *   | Wiki(PPL) | LA(PPL) |
> |-----------|--------|-------|-------|-------|-------|-------|-------|-------|-------|------|------|------|-----|--------|-----|------------|----------|
> | BF16      | 1T     | 61.52 | 68.71 | 66.54 | 38.14 | 69.33 | 76.33 | 24.54 | 33.81 | 8.2  | 34.99| 11.97| *   | 45.63 |  *   | 5.54       | 6.10     |
> | BF16      | 1.04T  | 61.46 | 68.57 | 64.48 | 38.91 | 70.09 | 75.41 | 24.73 | 30.94 | 8.6  | 34.91| 12.33|  *   | 45.13 |   *  | 5.54       | 6.00     |
> | FP4       | 1T     | 58.39 | 67.31 | 64.01 | 38.65 | 69.33 | 74.86 | 24.73 | 32.13 | 6.4  | 34.30| 12.09|   *  | 44.46 |  *   | 5.83       | 6.77     |
> | +QAF      | +40B   | 61.71 | 68.53 | 65.98 | 39.25 | 68.90 | 76.01 | 27.29 | 32.25 | 9.4  | 38.31|  12.11   |  *   | 45.75 |  *   | 5.56       | 5.97     |
>
> As the reviewer can see, the proposed method also generalizes for larger datasets. We will upload these results in the camera-ready version.
>
>
> $\textbf{Q4}:$ "No measured total BOPS, throughput, or average bit-width across FQT + QAF..."
>
> $\textbf{A4}:$ As stated in A3, we extend the pretraining phase to 1T tokens, followed by a 40B-token QAF phase. As noted in the general comments, during pretraining all three GEMMs operate in FP4, while during QAF only one of the three GEMMs uses FP4 (with the remaining two in BF16). The resulting average bit calculation is:$ [(1000\times3\times4) + (40\times1\times4) + (40\times2\times16)] / [1040\times3] = 4.3$ bits. We will include this result in the camera-ready version. Note the 5.2 bits result mentioned by the reviewer assumes fully 16bit QAF, which is wrong (see general comment above).
>
> $\textbf{Q5}:$ "Experiments rely on Gaudi2 simulation..."
>
> $\textbf{A5}:$ We agree with the reviewer that this is one of the limitations of our paper, as wrote in the limitation section. It is worth noting that even for Nvidia’s Blackwell, FP4 training is supported at the hardware level, but their software stack currently supports only inference. It is common for algorithmic research to precede full software support, and we expect that our work will encourage broader adoption of FP4 training across hardware accelerators. As stated in the limitations section, based on prior time-to-train improvements from BF16 to FP8 (35-40 \% time-to-train acceleration) and the additional 2× matrix multiplication speedup from FP8 to FP4, we provide a rough estimate of an 85\% reduction in training time.
>
> $\textbf{Q6}:$ "Stochastic-rounding implementation is unspecified..."
>
> $\textbf{A6}:$ The paper includes in the abstract anonymous full code reference to reproduce the experiments, specifically notice "experimental/fp4.py" line 6 which includes the stochastic rounding implementation. In general, the implementation includes taking the full precision value, adding random uniform noise with support equal to the bin size, then rounding to nearest quantization value:
> $Q(x) = \mathrm{round} (x+ U)$, where $x$ is full precision value, $U\sim [-0.5\delta,0.5\delta]$ is random uniform noise and $\delta$ is the local spacing between two representable FP4 values around x.
>
> $\textbf{Q7}:$ "Related-work survey omits two sub-FP8 methods..."
>
> $\textbf{A7}:$ Thank you for these references, we will add them in the camera ready version. Notice that, neither QuEST nor AdaBin represents fully quantized training. In these works, quantization is applied only to the forward general matrix multiplication (Equation 1), while the backward (Equation 2) and update (Equation 3) steps are kept in high precision. This significantly reduces the potential acceleration compared to our approach. As explicitly stated in QuEST [1], page 4: "although the backward computation is performed w.r.t. the quantized weights and activations, the multiplications and gradient operands are performed in standard 16-bit precision.".
>
> $\textbf{Q8}:$ "Training hyperparameters are incomplete..."
>
> $\textbf{A8}:$ We maintain all hyperparameters consistent with Llama2 paper, which includes the use AdamW optimizer with $\beta_1=0.9$, $\beta_2=0.95$. We use cosine learning rate schedule, with 2000 steps of warmup, peak learning rate of $3\times10^{-4}$ and decay to 0.1 of the peak lr. We use global batch-size of 4M tokens. All these details are available in the supplied code and will be added in the camera-ready version.
>
> $\textbf{Q9}:$ "All results are based on a single seed..."
>
> $\textbf{A9}:$ Table 3 includes running 0-shot downstream tasks on the final FP4 model and the BF16 baseline, which is a deterministic process (No stochastic process during inference). A possible way to get confidence bars is run all training again with different seed, but it is not feasible (full 1T tokens training take 30 days). In order to get an approximate confidence bar, we run the inference process presented in table 3 with 3 close checkpoints and calculate the std of these runs:
> | Precision | LA    | HS    | WG    | AC    | BQ    | PQ    | Wiki | LA |
> |-----------|-------|-------|-------|-------|-------|-------|---------|------|
> | BF16      | 1.27 | 0.13 | 0.89 | 0.82 | 0.66 | 0.45 | 0.016  | 0.2  |
> | FP4       | 0.33 | 0.15 | 1.36 | 0.39 | 0.76 | 0.25 | 0.02  | 0.04  |
>
> $\textbf{Q10}:$ "No empirical study on how long QAF must run..."
>
> $\textbf{A10}:$ Great question! We conducted an ablation study to determine how long QAF must run, i.e., how many tokens are required to reach the same loss as the BF16 baseline. Starting from three different checkpoints, we obtained the following results:
> * From 200B tokens – QAF requires 20B tokens (10%).
> * From 500B tokens – QAF requires 28B tokens (5.6%).
> * From 1T tokens – QAF requires 40B tokens (4%).
>
> As the reviewer can observe, for larger and real-world datasets, the QAF ratio decreases. Additionally, we experimented with different peak learning rates for QAF and concluded that it should be similar to the training learning rate—using a larger LR causes divergence, while a smaller LR prevents QAF from converging to the baseline. We plan to include this ablation study in the camera-ready version.
>
> $\textbf{Q11}:$ "No exploration of combining the method with PTQ for additional compression..."
>
> $\textbf{A11}:$ SmoothQuant and GPTQ are post-training quantization (PTQ) methods designed to take models trained in high precision and quantize them to lower precision (typically INT4) after training, while attempting to minimize quantization error for inference. As noted in the "general comment" above, our proposed pretraining + QAF approach produces a model directly in 4-bit precision (FP4), with no additional steps required to enable 4-bit inference. Therefore, we believe PTQ methods are not necessary in our proposed framework. We decided to focus on FP4 and not in INT4 since it is supported natively in modern hardware, such as Blackwell.
>
> $\textbf{Q12}:$ "Minor text issue: .."
>
> \textbf{A12}: Thank you, we will fix them in the camera-ready version

---

> > ### Comment · Reviewer_TyRD · 2025-08-05
> >
> > I want to thank the authors for their detailed rebuttal. My concerns have been sufficiently addressed, and the new experiments they've provided reinforce their findings. I have updated my score accordingly, and I believe this paper represents a significant breakthrough in the field.

---

### Official Review · Reviewer_34ZT · 2025-07-03

**Clarity:** 4
**Significance:** 4
**Originality:** 3
**Rating:** 5
**Confidence:** 5

**Summary:**

This paper explores FP4 training of LLMs using NVFP4 format. The authors use stochastic rounding for the backward path, which is instrumental in getting unbiased gradients. Furthermore, the authors provide theoretical and empirical analysis of the quantized SGD convergence. The proposed method also explores switching to higher precision training when quantization noise dominates gradient updates. The paper is well structured, well-written, and provides insightful information.

**Questions:**

As mentioned in the previous section:

1. More evaluation results such as GPQA, IFEveal ( Open LLM Leaderboard 2 ), EvalPlus will add more insight on how well the quantized model is trained.

2. Did authors use any sort of Maximal Update Parametrization (muP)? What was the effect of quantization on that? [1]

3. Is it possible to derive some mathematical guarantee that quantized SGD converges given that higher precision SGD converges? Example: refer to Section 4 of [2]

References:

[1] Yang, Greg, et al. "Tensor programs v: Tuning large neural networks via zero-shot hyperparameter transfer." arXiv preprint arXiv:2203.03466 (2022).

[2] Ghaffari, A, et al. "Is integer arithmetic enough for deep learning training?." Advances in Neural Information Processing Systems 35 (2022): 27402-27413.

**Ethical Concerns:**

["NO or VERY MINOR ethics concerns only"]

**Limitations:**

Authors adequately addressed the limitation of this work.

**Paper Formatting Concerns:**

Paper is well written and well formatted,

**Quality:**

3

**Strengths And Weaknesses:**

**Strengths**
1. This paper is proposing a solid methodology and mathematical derivation. I personally enjoyed seeing mathematical derivation accompanied with empirical evidence,

2. Fig 1 and Fig 2  provide insightful intuition regarding the number formats. Also Fig 4 and Fig 5 are informative and provide empirical evidence for mathematical derivation of critical quantization noise level.

3. Fig 6 shows quantized training loss is similar to the baseline training, showing the effectiveness of this algorithm for Llama 7B model.

4. Experimental evidence with perplexity and zero shot tasks shows strong evidence that training procedure was effective.

**Weaknesses**

This paper does not have too many weaknesses; here are my suggestions to further improve the quality of the presentation:

1. As mentioned by the authors, the lack of validation in FP4 hardware is a key limitation of this paper.

2.  It would also be insightful to see the results of FP4 training for more models and more evaluation tasks such as EvalPlus and GPQA.

3. Section 4.1 can include quantized SGD convergence analysis to make sure and understand the conditions of quantized SGD convergence.

---

> ### Author Rebuttal · Authors · 2025-07-31
>
> $\textbf{Q1}:$ "As mentioned by the authors, the lack of validation in FP4 hardware is a key limitation of this paper."
>
> $\textbf{A1}:$ We agree with the reviewer that this is one of the limitations of our paper. It is worth noting that, even for Nvidia’s Blackwell, FP4 training is supported at the hardware level, but their software stack currently supports only inference. It is common for algorithmic research to precede full software support, and we expect that our work will encourage broader adoption of FP4 training across hardware accelerators.
>
> $\textbf{Q2}:$ "More evaluation results such as GPQA, IFEveal ( Open LLM Leaderboard 2 ), EvalPlus will add more insight on how well the quantized model is trained."
>
> $\textbf{A2}:$ We have extended the primary training of the 7B model (Figure 6) to 1 trillion tokens, followed by 40B tokens of QAF, which requires approximately 30 days on 256 devices. We achieved training loss of 1.752 (FP4) Vs 1.76 (BF16). Then we compare the BF16 baseline  with FP4 model on downstream tasks (similar to Table 3), including new tasks as required by the reviewer - GPQA (GQ), IfEval (IF), MBPP (MB), TrivialQA (TQ), XSum (XS):
>
> | Precision | Tokens | LA    | HS    | WG    | AC    | BQ    | PQ    | GQ    | IE    | MB   | TQ   | XS   |  *   | Avg   |  *   | Wiki(PPL) | LA(PPL) |
> |-----------|--------|-------|-------|-------|-------|-------|-------|-------|-------|------|------|------|-----|--------|-----|------------|----------|
> | BF16      | 1T     | 61.52 | 68.71 | 66.54 | 38.14 | 69.33 | 76.33 | 24.54 | 33.81 | 8.2  | 34.99| 11.97| *   | 45.63 |  *   | 5.54       | 6.10     |
> | BF16      | 1.04T  | 61.46 | 68.57 | 64.48 | 38.91 | 70.09 | 75.41 | 24.73 | 30.94 | 8.6  | 34.91| 12.33|  *   | 45.13 |   *  | 5.54       | 6.00     |
> | FP4       | 1T     | 58.39 | 67.31 | 64.01 | 38.65 | 69.33 | 74.86 | 24.73 | 32.13 | 6.4  | 34.30| 12.09|   *  | 44.46 |  *   | 5.83       | 6.77     |
> | +QAF      | +40B   | 61.71 | 68.53 | 65.98 | 39.25 | 68.90 | 76.01 | 27.29 | 32.25 | 9.4  | 38.31|  12.11   |  *   | 45.75 |  *   | 5.56       | 5.97     |
>
>
> $\textbf{Q3}:$ "Section 4.1 can include quantized SGD convergence analysis to make sure and understand the conditions of quantized SGD convergence.
> Is it possible to derive some mathematical guarantee that quantized SGD converges given that higher precision SGD converges? Example: refer to Section 4 of [2]"
>
>
> $\textbf{A3}:$ Thank you for this interesting reference. The global convergence guarantee in [2] and our local analysis in Section 4.1 work together. The theory in [2] predicts that training will eventually stall in a noise-dependent neighborhood, while our diagnostic tool helps identify in real-time when that stalling has begun.
>
>
> $\textbf{Q4}:$ "Did authors use any sort of Maximal Update Parametrization (muP)? What was the effect of quantization on that? [1]"
>
> $\textbf{A4}:$ We thank the reviewer for this valuable suggestion. In our work, the primary goal was to demonstrate that full FP4 training can achieve convergence with results on par with the BF16 baseline, using the same hyperparameters. This means we did not explore techniques such as muP, but we consider this an interesting direction for future work. Notice, as mentioned in A2, the main experiment (1T tokens) requires approximately 30 days on 256 devices.

---

### Official Review · Reviewer_hLJD · 2025-07-04

**Clarity:** 3
**Significance:** 3
**Originality:** 3
**Rating:** 5
**Confidence:** 2

**Summary:**

This paper presents the first successful demonstration of FP4 training for weights, activations, and gradients. Leveraging the NVFP4 format and a carefully selected rounding strategy (stochastic rounding for backward and round-to-nearest for forward), the authors train a 7B LLaMA2 on 200B tokens using 256 Intel Gaudi2 devices. The trained FP4 model achieves performance on par with BF16 after a brief quantization-aware finetuning (QAF) phase.

**Questions:**

Could the authors clarify the scope and applicability of the proposed FP4 training method? Specifically, are there particular model types, sizes, or tasks for which this approach is especially suitable or advantageous? Or do the authors believe it can generalize effectively to any LLM training scenario?

**Ethical Concerns:**

["NO or VERY MINOR ethics concerns only"]

**Final Justification:**

This is a solid paper that I recommend for acceptance.

**Limitations:**

Yes

**Quality:**

3

**Strengths And Weaknesses:**

Strengths

- The paper presents the first demonstration of FP4 across all three GEMMs (forward, backward, update) at scale (7B model, 200B tokens).
- The use of FP4 for both weights and activations has the potential to largely accelerate training.
- The theoretical analysis of the gradient-to-noise threshold—supported by empirical evidence—offers valuable insight into when higher-precision training should be resumed.
- It is great to see the open-sourced code.

Weaknesses

While large-scale experiments are understandably expensive, it would strengthen the paper to include broader evaluations.

- It would be great to evaluate your methods across different model sizes or architectures.
- A key limitation is that the proposed FP4 training is only simulated on Gaudi2 hardware, which lacks native FP4 support. As such, the actual speed and memory benefits remain speculative. It would be great if you could provide a solid prediction for the cost reduction.
- The paper lacks qualitative or in-depth comparison with other recent FP4 training approaches. The baseline is only BF16.
- Details of the training configuration—such as optimizer type, learning rate schedule, batch size, and initialization—are only briefly mentioned and could be more thoroughly documented for full reproducibility.

---

> ### Author Rebuttal · Authors · 2025-07-31
>
> $\textbf{Q1}:$ "It would be great to evaluate your methods across different model sizes or architectures"
>
> $\textbf{A1}:$ In the camera-ready version, we will include additional experiments involving smaller models with 125M and 350M parameters, achieved final on-par training loss with the BF16 baseline (125M: 2.922(BF16) Vs 2.918 (FP4). 350M: 2.67 (BF16) Vs 2.673 (FP4)). Furthermore, we have extended the primary training of the 7B model (Figure 6) to 1 trillion tokens, followed by 40B tokens of QAF, which requires approximately 30 days on 256 devices. We achieved training loss of 1.752 (FP4) Vs 1.76 (BF16). Then we compare the BF16 baseline  with FP4 model on downstream tasks (similar to Table 3), including new tasks - GPQA (GQ), IfEval (IF), MBPP (MB), TrivialQA (TQ), XSum (XS):
>
> | Precision | Tokens | LA    | HS    | WG    | AC    | BQ    | PQ    | GQ    | IE    | MB   | TQ   | XS   |  *   | Avg   |  *   | Wiki(PPL) | LA(PPL) |
> |-----------|--------|-------|-------|-------|-------|-------|-------|-------|-------|------|------|------|-----|--------|-----|------------|----------|
> | BF16      | 1T     | 61.52 | 68.71 | 66.54 | 38.14 | 69.33 | 76.33 | 24.54 | 33.81 | 8.2  | 34.99| 11.97| *   | 45.63 |  *   | 5.54       | 6.10     |
> | BF16      | 1.04T  | 61.46 | 68.57 | 64.48 | 38.91 | 70.09 | 75.41 | 24.73 | 30.94 | 8.6  | 34.91| 12.33|  *   | 45.13 |   *  | 5.54       | 6.00     |
> | FP4       | 1T     | 58.39 | 67.31 | 64.01 | 38.65 | 69.33 | 74.86 | 24.73 | 32.13 | 6.4  | 34.30| 12.09|   *  | 44.46 |  *   | 5.83       | 6.77     |
> | +QAF      | +40B   | 61.71 | 68.53 | 65.98 | 39.25 | 68.90 | 76.01 | 27.29 | 32.25 | 9.4  | 38.31|  12.11   |  *   | 45.75 |  *   | 5.56       | 5.97     |
>
> The Scaling to larger model sizes is not feasible with our current resources. Our experiments are based on the LLaMA 2 architecture, which is widely adopted in most LLMs.
>
> $\textbf{Q2}:$ "A key limitation is that the proposed FP4 training is only simulated on Gaudi2 hardware, which lacks native FP4 support. As such, the actual speed and memory benefits remain speculative. It would be great if you could provide a solid prediction for the cost reduction."
>
> $\textbf{A2}:$ We agree with the reviewer that this is one of the limitations of our paper (as we wrote in our limitation section). It is worth noting that even for Nvidia’s Blackwell, FP4 training is supported at the hardware level, but their software stack currently supports only inference. It is common for algorithmic research to precede full software support, and we expect that our work will encourage broader adoption of FP4 training across hardware accelerators. As stated in the limitations section, based on prior time-to-train improvements from BF16 to FP8 (35-40 \% time-to-train acceleration) and the additional 2× matrix multiplication speedup from FP8 to FP4, we provide a rough estimate of an 85\% reduction in training time.
>
> $\textbf{Q3}:$ "The paper lacks qualitative or in-depth comparison with other recent FP4 training approaches. The baseline is only BF16."
>
> $\textbf{A3}:$ Our paper is the first and only work to demonstrate full FP4 training, where all three general matrix multiplications (GEMMs) described in Equations 1, 2, and 3 are quantized. Therefore, there is no prior work that allows for a fair comparison. The most related studies, summarized in Table 2, apply only partial quantization (offering less potential acceleration) and are conducted on smaller datasets.
>
> $\textbf{Q4}:$ "Details of the training configuration—such as optimizer type, learning rate schedule, batch size, and initialization—are only briefly mentioned and could be more thoroughly documented for full reproducibility."
>
> $\textbf{A4}:$ We maintain all hyperparameters consistent with Llama2 paper, which includes the use AdamW optimizer with $\beta_1=0.9$, $\beta_2=0.95$. We use cosine learning rate schedule, with 2000 steps of warmup, peak learning rate of $3\times10^{-4}$ and decay to 0.1 of the peak lr. We use global batch-size of 4M tokens. All these details are available in the supplied code and will be added in the camera-ready version.
>
>
> $\textbf{Q5}:$
> "Could the authors clarify the scope and applicability of the proposed FP4 training method? Specifically, are there particular model types, sizes, or tasks for which this approach is especially suitable or advantageous? Or do the authors believe it can generalize effectively to any LLM training scenario?"
>
> $\textbf{A5}:$ The primary applicability of the proposed FP4 training is to accelerate the pretraining phase, which is the most computationally expensive step and becomes increasingly demanding with larger datasets and models. Our work focuses on the LLaMA architecture, which is one of the most widely used architectures in modern LLMs, and we believe our approach can be effectively extended to most LLM models. While vision transformers were not the focus of this work, their similar architectural structure suggests that our method could also provide acceleration in that domain.

---

> > ### Comment · Reviewer_hLJD · 2025-08-06
> >
> > Thank you for addressing my concerns. I am satisfied with the authors’ responses and will maintain my original acceptance score.

---

### Comment · Area_Chair_xpH3 · 2025-08-03
**Reminder: Discussion Phase (July 31 – Aug 6)**

Hi everyone,

This is a reminder that the discussion phase is between July 31 – Aug 6.

Please read the author responses, especially where you are mentioned, and post your reply as soon as possible. This helps ensure there's time for meaningful back-and-forth.

Thanks for your engagement!

AC

---

### Decision · Program_Chairs · 2025-09-17

**Decision:**

Accept (spotlight)

**Comment:**

This paper presents a practical approach to training large language models using FP4 format, along with comprehensive evaluations of various components (e.g., rounding functions and scaling formats). The research effectively bridges theory and practice by providing theoretical justifications supported by empirical evidence. All reviewers are clear to accept this paper, and the authors adequately addressed reviewer concerns during the rebuttal period. This timely work delivers remarkable results on an important research direction. I believe it will provide valuable guidance to the community for low-precision training of LLMs.